# Botulinum Toxin Type A for Treatment of Forehead Hyperhidrosis: Multicenter Clinical Experience and Review from Literature

**DOI:** 10.3390/toxins14060372

**Published:** 2022-05-27

**Authors:** Anna Campanati, Emanuela Martina, Stamatis Gregoriou, George Kontochristopoulos, Matteo Paolinelli, Federico Diotallevi, Giulia Radi, Ivan Bobyr, Barbara Marconi, Giulio Gualdi, Paolo Amerio, Annamaria Offidani

**Affiliations:** 1Dermatological Clinic, Polytechnic Marche University, 60200 Ancona, Italy; ema.martina@gmail.com (E.M.); matteopaolinelli.an@gmail.com (M.P.); federico.diotallevi@gmail.com (F.D.); radigiu1@gmail.com (G.R.); ivan.bobyr@ospedaliriuniti.marche.it (I.B.); barbara.marconi@ospedaliriuniti.marche.it (B.M.); a.offidami@ospedaliriuniti.marche.it (A.O.); 2First Department of Dermatology-Venereology, Faculty of Medicine, National and Kapodistrian University of Athens, “Andreas Sygros” Hospital for Cutaneous and Venereal Diseases, 16121 Athens, Greece; stamgreg@yahoo.gr; 3Departments of Dermatology-Venereology Andreas Sygros Hospital, 16121 Athens, Greece; g.kontochristopoulos@yahoo.gr; 4Dermatologic Clinic, Department of Medicine, and Aging Science, University D’Annunzio Chieti-Pescara, 66100 Chieti, Italy; giulio.gualdi@unich.it (G.G.); p.amerio@unich.it (P.A.)

**Keywords:** forehead hyperhidrosis, Incobotulinum toxin type A, quality of life, efficacy, safety, HDSS, DLQI, gravimetric test

## Abstract

Among the forms of idiopathic hyperhidrosis, those involving the forehead have the greatest impact on patients’ quality of life, as symptoms are not very controllable and are difficult to mask for patients. Although the local injection therapy with Incobotulinum toxin type A (IncoBTX-A therapy) can be considered a rational treatment, data from the literature describing both efficacy and safety of the treatment over the long term are poor. The aim of this report is to describe the single-center experience of five patients seeking treatment, for forehead hyperhidrosis with IncoBTX-A. To evaluate the benefits, safety profile and duration of anhidrosis, patients were treated following a standardized procedure and then followed until clinical relapse. The amount of sweating was measured by gravimetric testing, the extension of hyperhidrosis area was measured through Minor’s iodine starch test, and response to the treatment was evaluated using the Hyperhidrosis Disease Severity Scale (HDSS) and the Dermatology Life Quality Index (DLQI). In all treated patients, a significant anhidrotic effect was observed 4 weeks after the treatment and lasted for approximately 36 weeks. The reduction in sweat production was associated with significant amelioration of symptoms and quality of life for all treated patients. No serious side effects occurred; one patient complained of a mild transient bilateral ptosis. Although further wider studies are required, our preliminary results seem to encourage the use of IncoBTX-A in forehead hyperhidrosis.

## 1. Introduction

Primary focal hyperhidrosis commonly affects axillae, palms, and soles, but may occur on many body sites, including head, sub-mammary regions, and groin [1].

Primary focal head hyperhidrosis (HH) most commonly displays one of the four following patterns of involvement: the forehead; a band-like distribution around the scalp, known as the ophiasis pattern; a combination of the forehead and ophiasis scalp; or the entire scalp and forehead. Several areas can be involved simultaneously in the same patient [2].

Following a step-wise treatment approach in patients with HH, botulinum toxin type A (BTX-A) seems to be a reasonable treatment for patients unresponsive to topical antiperspirants, for its proven efficacy and safety in most hyperhidrotic areas of the body [3,4]. 

In the literature, few data have been reported on the use of three different types of BTX-A for treating HH: Onabotulinum toxin type A (OnaBTX-A: BOTOX ^®^; Allergan, Inc., Irvine, CA, USA), Abototulinum toxin type A (AboBTX-A: DYSPORT ^®^; Ipsen Ltd., Slough, UK) and Prabobotulinum toxin type A (PraBTX-A: NABOTA ^®^, Daewoong, Co., Ltd., Seoul, Korea). However, no data have been reported on the use of Incobotulinum toxin type A (IncoBTX-A, XEOMIN ^®^, Merz Pharmaceuticals GmbH GmbH & Co KGaA, Frankfurt, Germany) in HH IncoBTX-A for the management of patents suffering from HH.

Some studies have already demonstrated IncoBTX-A is effective and safe for the treatment of primary axillary hyperhidrosis [5,6] and some authors failed to demonstrate any difference in efficacy and safety, when IncoBTX-A and OnaBTX-A were compared for treatment in this area [7,8]. Moreover, IncoBTX-A has been successfully used for the treatment of palmar hyperhidrosis [9].

The aim of this manuscript is to report our clinical experience related to the use of IncoBTX-A in patients suffering from HH. 

## 2. Results

All treated patients responded to an IncoBTX-A injection (Figure 1).

### 2.1. IncoBTX-A Effect on Sweat Production

The treatment with IncoBTX-A produced a decrease in sweat production in all treated patients, and mean sweat amount significantly decreased from a baseline value of 0.184 ± 0.02 gr/min to a post-treatment value of 0.11 ± 0.02 gr/15 min (*p* = 0.009) (Table 1; Figure 2). 

### 2.2. IncoBTX-A on Extent of Hyperhidrotic Area 

The extent of hyperhidrotic area, measured through Minor’s iodine starch test, decreased dramatically after injection of IncoBTX-A, varying from a mean pretreatment value of 16.4 ± 2.7 to a mean post-treatment value of 0.4 ± 0.54 (*p* = 0.004). In the following weeks, patients reported a gradual decrease in anhidrotic effect, with a progressive increase in minor’s iodine starch test value, with restoration of the basal condition 36 weeks after treatment (Table 1; Figure 3).

### 2.3. Patients’ Clinical Improvement 

A decrease in sweat production was confirmed by patients’ clinical response. All patients were responsive to the treatments, reporting a 2-point improvement in HDSS, and HDSS mean values ± DS dramatically decreased from baseline (3.6 ± 0.54) to T4 (0.4 ± 0.54) in all treated patients (*p* = 0.001). After 12 weeks of treatment, one patient relapsed, after 20 weeks, three patients, four patients after 28 weeks, and after 36, all five patients relapsed (Table 1; Figure 4).

### 2.4. Patients’ Quality of Life

DLQI dramatically decreased from mean baseline value of 24.6 ± 2.7 to a post treatment value of 1.4 ± 0.54 (*p* = 0.0007). In the following weeks, with the reduction in the anhidrotic effect, the quality of life of the patients returned to rise, without returning to pre-treatment levels, even at 36 weeks, despite the relapse of all patients (Table 1; Figure 5). 

### 2.5. Safety Profile

None among the treated patients experienced serious side effects. All patients reported pain during injection, all patients complained of pain during treatment reported as “bearable” and lacking any deterrent effects on further future treatments. No compensatory hyperhidrosis was observed among treated patients. 

## 3. Discussion

Head hyperhidrosis is associated with a great impact on a patient’s quality of life, above all because it is poorly responsive to topical therapy or oral anticholinergics and cannot benefit from surgical treatment that would lead to difficult-to-treat aesthetic-functional scarring [10,11]. Moreover, it is more difficult to hide for patients compared to hyperhidrosis involving other body areas [12].

BTX-A is a safe and effective drug for treating focal hyperhidrosis, providing longer-lasting results than topical treatments, without the necessity of invasive surgical procedures [13,14,15,16,17]. Although there are no guidelines supported by scientific societies for the treatment of craniofacial hyperhidrosis, the International Hyperhidrosis Society places botulinum toxin type A in the second line of treatment in craniofacial hyperhidrosis [18]. However, there are only very few literature reports on the efficacy and safety of use of BTX-A in craniofacial hyperhidrosis, and none of them specifically related to IncoBTX-A (Table 2).

Boger et al. firstly, in 2000, reported their successful experience in the use of BTX-A for the treatment of craniofacial hyperhidrosis in 12 patients. After confirming the diagnosis by Minor’s iodine starch test, authors treated one-half of the forehead with an injection of 2.5–4 ng AboBTX-A equidistantly intracutaneously. After 4 weeks, they assessed the efficacy using Minor’s iodine starch control test and then treated the other half. Another 4 weeks later, a standardized telephone interview was carried out. After 1–7 days, the craniofacial sweating in the injected area had completely ceased in 11 patients and was mildly reduced in the remaining one. The efficacy was confirmed by repeated Minor’s iodine starch tests. Mild weakness of frowning was the only side effect, lasting 1–12 weeks and completely resolving in all patients. Although sweating has not yet recurred in most patients at follow-up periods up to 27 months, one patient had a relapse 9 months after treatment [19]. 

These results were also confirmed by Kinkelin I, et al. [20] in 2000, who treated 10 male patients suffering from frontal hyperhidrosis with OnaBTX-A, injected intracutaneously at a mean dosage of 86 mouse units. The treatment provided a significant reduction in sweat 4 weeks after treatment, and the effect lasted at least 5 months in 9 of the 10 patients. All patients subjectively judged the treatment as very effective. Minor side effects included painful injections and a transient weakness in the forehead muscles without ptosis. 

In 2013, George SMC et al. [21] reported their successful experience with the treatment of craniofacial hyperhidrosis with OnaBTX-A, and they emphasized how important it is for dermatologists to be aware of the application of botulinum toxin and the practical aspects of treatment for managing such patients [16]. 

In 2014, Ko EJ et al. [22] compared the efficacy and diffusion of three formulations of botulinum toxin type A in two patients with forehead hyperhidrosis, by injecting onabotulinm toxin type A in both patients with the three different formulations on the right (3.3 U OnaBTX-A), middle (8.3 U AboBTX-A) and left (3.3 U PraBTX-A) areas of the forehead. The authors reported that BTX-A injections proved to be an effective treatment for forehead hyperhidrosis. The amount of sweating was markedly reduced. The different areas of anhidrotic halo support the suggestions in previous reports [23,24] that the area of diffusion appears to be greater with AboBTX-A than with OnaBTX-A and PraBTX-A. 

**Table 2 toxins-14-00372-t002:** Literature reports on the efficacy and safety of use of BTX-A in craniofacial hyperhidrosis.

Reference	Patients	Type of BTX-A	Outcome and Methods	Results	AE
**Böger et al.** [17]	12	**AboBTX-A** injected in one-half of the forehead at the dosage of 2.5–4 ng	Effectiveness on frontal hyperidrosis assessed at T0 and after 4 weeks by - Minor’s iodine starch test	After 1–7 days the craniofacial sweating in the area injected had completely ceased in 11 patients and was mildly reduced in the remaining one. One patient had a relapse 9 months after treatment.	Mild weakness of frowning
**Kinkelin et al.** [18]	10	**OnaBTX-A**, injected intracutaneously at mean dosage of 86 mouse units	Effectiveness on frontal hyperidrosis assessed at T0 and after 4 weeks by: - Minor’s iodine-starch test - Gravimetric assessment - Photograph	Reduction of sweat after 4 weeks treatment, lasted at 5 months in nine of the 10 patients	Painful injections and a transient weakness of forehead muscles without ptosis
**George et al.** [19]	4	**OnaBTX-A** injected using a dose of 1–2 units per injection	Effectiveness on rarer forms of focal facial hyperidrosis (upper lip and chin, both cheeks, central face and frontal scalp respectively) assessed at T0 and after 6 weeks by Minor’s iodine-starch test	Reduction of sweat, lasted at mean 6–8 months after treatment.	Patient with frontal hyperhidrosis: loss of rhytides on her forehead and mild brow ptosis; Patient with perioral sweating: droop of his left upper lip.
**Ko EJ et al.** [20]	2	**OnaBTX-A** injected on the right of forehead (3.3 U), **AboBTX-A** injected on the middle (8.3 U) **PraBTX-A** injected on the left (3.3 U) All injections were of identical volume (0.1 mL).	Comparison of the efficacy and diffusion of three formulations of botulinum toxin type A in three different areas of the forehead by the following assesments at T0 and after 2 weeks: - Minor iodine–starch test - Transepidermal water loss (TEWL) - Corneometer	The area of diffusion appears to be greater with AboBTX-A covering 6.7% of the total area, while OnaBTX-A and PraBTX-A produced similarly sized areas of anhidrosis (2.5% and 2.6%, respectively).	Minimizing the area of diffusion is important to minimize the potential for adverse effects (AEs).
**Ando Y et al.** [23]	15	Not specified	Effectiveness on frontal hyperidrosis assessed at T0 and after 4 weeks by - Ventilation capsule method - Minor’s iodine-starch test. - HDSS - DLQI	Remarkable antiperspirants effect observed from 2 weeks after injection, and the effect last for approximately 30 weeks.	Two patients complained of transient mild ptosis

Ando Y et al. [25], in 2021, reported their experience to clarify the benefits of BTX-A for HFH in 15 patients. In most cases, a remarkable antiperspirant effect was observed from 2 weeks after injection, and the effect lasted for approximately 30 weeks. Two patients complained of transient mild ptosis but no other serious side effects were reported. The authors did not report what type of BTX-a was used in their case series [25]. 

Nowadays, treatment of HH is far from standardization, above all because it is a little-practiced treatment, not only for its cost, but mainly because of cultural deficiencies among dermatologists on the appropriateness of treatments for hyperhidrosis. Several aspects, such as the administration method, dose, period of suppressed sweating, and safety, have not been sufficiently investigated. 

Our clinical experience, which represents the first one related to the use of IncoBTX-A on five patients with HH, fits into this open scenario, with the aim of providing additional clinical experience, including long-term follow up. 

IncoBTX-A is a molecule with at least two potential advantages compared to AboBTX-A and OnaBTX-; it is more manageable owing to its stability at room temperature, and it is less immunogenic, owing to the lack of complexing proteins [5,26,27,28,29].

However, successful treatment is mainly based on the identification of the optimal diffusion area of the BTX-A used, which is the result of a subtle balance influenced by the drug formulation, the dose, the injection volume, and the injection site [28].

The greater the diffusion area, the more likely it is that adverse events will occur, and the more difficult it is to ensure accurate localization. Minimizing the area of diffusion is crucial to reduce the potential for adverse effects, especially when injection sites are close to other muscles, as is the case when treating the forehead. However, clinicians also need to avoid the diffusion area being too small, otherwise there is no effect on hyperhidrosis. 

Due to the lower molecular weight of IncoBTX-A, related to the lack of complexing proteins, it has been hypothesized that it may diffuse more to the surrounding tissues than other BTX-A [27], with potential greater impact on muscle strength, although our preliminary results did not confirm this risk in the treatment of palms [30]. 

Nevertheless, IncoBTX-A also seems to confirm its safety and efficacy in the treatment of HH. Sweat production dramatically decreased after administration of BTX-A (Figure 1), the anhidrotic effect resulted in the disappearance of the hyperhidrotic area (Figure 2), and a significant improvement in disease severity (Figure 3) and patients’ quality of life was found 4 weeks after the treatment (Figure 4). The sweat suppression period lasted up to 36 weeks; this result is in line with the mechanism of action of IncoBTX-A, which can induce functional chemo-denervation of cholinergic nerve terminations, through molecular disruption of the SNAP-25 protein. The duration of the anhidrosis phase varies according to the individual resynthesis ability of SNAP 25, and in the frontal region, data from the literature agree with the length of time observed by us.

Interestingly, patients’ quality of life remained improved compared to prior to treatment, even when efficacy was determined to be lost by clinometric indices (HDSS, Minor’s and gravimetric tests). This evidence probably reflects an innovative trust of patients towards treatment, as a solution to a problem that is no longer perceived as incurable. Although treatment is perceived as mildly uncomfortable by patients, owing to pain perceived during injections, they all declare that they are willing to repeat the treatment at the time of loss of efficacy. This propensity for treatment is also related to the absence of severe side effects in our series of patients. Only one patient reported mild bilateral ptosis, completely resolved in 12 weeks. Ptosis may have been favored by the rubbing of the skin of the forehead that the patient has recklessly practiced removing the marks made to identify the grid for the injections. Rubbing of the skin could have promoted migration of BTX-A toward muscular layers, favoring its interaction with neuromuscular junctions.

Although further studies on larger series of patients need to be performed, our preliminary results seem to agree with those reported by other authors for the treatment of HH with other types of BTX-A, including OnaBTX-A, AboBTX-A, and PraBTX-A. 

## 4. Materials and Methods

### 4.1. Patients

This is a case series including 5 patients affected by head hyperhidrosis and treated with IncoBTX-A after written informed consent was obtained prior to beginning specific procedures. 

The study was conducted according to the Helsinki Declaration ad after ethical approval from the Ethics Committee of Azienda Ospedaliero-Universitaria Ospedali Riuniti di Ancona (Protocol 2008 0710 OR on 28 Jan 2008.

Patients (3 female and 2 males, mean age 42.3 ± 7 years, age range 29–52) were suffering from moderate to severe primary head hyperhidrosis with forehead pattern of presentation resistant to antiperspirants containing aluminium chloride (resistance to treatment was defined as less than two-points improvement in HDSS from baseline value). None among them had received BTX-A treatment for any reasons in the last 12 months or had been treated with antiperspirants containing aluminum chloride or iontophoresis less than 3 months before (Table 1). Informed consent was obtained from all subjects involved in the study. All patients underwent a complete evaluation consisting of clinical assessment of hyperhidrotic area. The clinical assessment included a baseline, and post-treatment examination of hyperhidrotic patients, starting from four weeks after treatment and every 8 weeks until relapse. It was based on classification of disease severity (Hyperhidrosis Diseases Severity Scale—HDSS), quantification of sweat production (gravimetric test), identification of hyperhidrotic area extension (minor’s iodine starch test) and evaluation of patients’ quality of life (Dermatology Life Quality Index—DLQI).

### 4.2. Classification of Hyperhidrosis Severity 

Disease severity was evaluated through the Hyperhidrosis Disease Severity Scale (HDSS) [31]. The Hyperhidrosis Disease Severity Scale (HDSS) is a disease-specific, quick, easily understood and validated diagnostic tool that provides a qualitative measure of the severity of the patient’s condition based on how it affects daily activities. Patients were asked to rate their hyperhidrosis by selecting the statement that best reflects his or her experience with forehead sweating, considering a specific score for each response as follows: “My sweating is never noticeable and never interferes with my daily activities” score 1; “My sweating is tolerable but sometimes interferes with my daily activities” score 2; “My sweating is barely tolerable and frequently interferes with my daily activities” score 3; “My sweating is intolerable and always interferes with my daily activities” score 4. A score of 3 or 4 indicated severe hyperhidrosis, and a score of 1 or 2, mild or moderate hyperhidrosis. A successful treatment was identified as an improvement from a score of 4 or 3 to a score 2 or 1, since a 1-point improvement in HDSS score is associated with a 50% reduction in sweat production and a 2-point improvement with an 80% reduction, according to the Canadian Hyperhidrosis Advisory Committee, and relapse of hyperhidrosis was identified as 1-point worsening in HDSS score after treatment (50% increase in sweat production) [32]. 

### 4.3. Quantification of Sweat Production

Quantitative gravimetric measurement of sweat secretion was conducted with standardized filter paper (Melitta GmbH, Minden, Germany), which was preliminarily weighed on a high-precision laboratory scale (Sartorius, Hamburg, Germany, precision+ 0.5 mg). The paper was then applied to the forehead of the patients for exactly 15 min and weighed again, yielding the rate of sweat secretion in grams per 15 min. Since sweat production is influenced by room temperature and humidity, all data were collected under standardized environmental conditions: room temperature ranged from 20 to 22 °C, and ambient humidity from 55 to 60%. 

### 4.4. Identification of Hyperhidrotic Area Extension

The hyperhidrosis extension was evaluated by Minor’s iodine starch test, which is a common, safe, and effective test used in clinical practice to evaluate the extent of hyperhidrotic area. Patient’s skin was preliminarily dried with a 70% alcohol solution, then an iodine solution was applied and allowed to dry completely. Next, the skin was dusted with a thin film of starch (corn flour). Sweating was expected by leaving the patient at room temperature for 15 min. The patient’s hyperhidrotic area readily turned a dark blue shade when in contact with sweat, while the euhidrotic area remained unchanged. After that each forehead was divided in four areas and for each area we gave a score ranging from 0 to 5 according to color parameters given by Minor’s test: 0 for anhidrotic area or minimal sweating, 1 for initial or discrete sweating, 2 for mild sweating, 3 for moderate sweating, 4 for intense sweating, and 5 for oversweating, in order to obtain a global score ranging from 0 to 20 [33]. 

### 4.5. Patients’ Quality-of-Life Evaluation

Subjective evaluation of disability caused by the symptoms of hyperhidrosis was obtained using the Dermatology Life Quality Index (DLQI) which is a validated tool and demonstrates the degree of change following treatment of skin disease. It consists of 10 questions regarding work, leisure, daily activities, personal relationships, and treatments. Each question has five alternative answers: “very much”, “a lot”, “a little”, “not at all” or “not relevant” with corresponding scores of 3, 2, 1, 0, respectively (the answer “not relevant” is scored as 0, as suggested by Finlay et al. [34]). The results can be easily computed by summing the score of each question, resulting in a global value ranging from 0 to 30. The higher the score, the greater the impairment of quality of life. 

### 4.6. Treatment with IncoBTX-A

Treatment protocol was in accordance with suggestions provided by Ando Y et al. 2021 to treat forehead [25]. All patients received IncoBTX-A at a fixed dosage per cm^2^. A nurse external to the study diluted every vial of lyophilised IncoBTX-A containing 100 U (Xeomin^®^, Merz Pharma, GmbH & Co KGaA) in 5 mL sterile 0.9% saline solution. Following this dilution procedure, 1 mL of reconstituted product contained 20 U of IncoBTX-A, and 0.10 mL contained 2U.

A reference grid with square areas of 2.25 cm^2^ was drawn on the forehead; the intra-cutaneous injection of IncoBTX-A 0.10 mL (2U) was given by the physician in the central part of each square. Injections were done using a 30 G × 0.30 × 4 mm gauge needle, which was not replaced during the treatment. Injections were placed at least 2–3 cm above the eyebrows to avoid ptosis. Total injected dose of incoBTX-A per forehead ranged from 40 to 80 Mouse Units, depending on the size of the area. No local anesthesia was required for treatment. 

### 4.7. Statistical Analyses

The Graph-Pad Prism software (version 5.3, El Camino REAL, San Diego, CA, USA) was used to perform all statistical analyses. All data were continuous variables expressed as means ± SD. The normal distribution of continuous variables was verified with Kolmogorov–Smirnov test. Homogeneity of variance was tested by Cochran C, and post hoc comparison with a non-parametric test (Mann–Whitney U test) was used to discriminate between means of values. Levels of significance were set at *p* < 0.05.

Limit of the study: a modest number of patients were recruited. A more extensive case description is required.

## Figures and Tables

**Figure 1 toxins-14-00372-f001:**
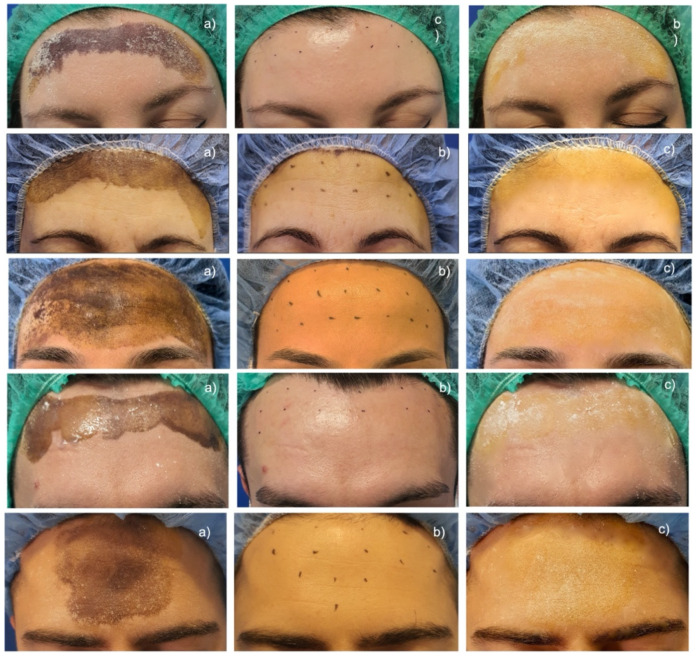
Effect of IncoBTX-A injection on forehead hyperhidrosis on five patients (**a**) Minor’s iodine starch test in all patients at baseline; (**b**) configuration of the pre-operative treatment grid in all patients; (**c**) effect of treatment on sweat production documented through minor’s iodine starch test 4 weeks after treatment in all treated patients.

**Figure 2 toxins-14-00372-f002:**
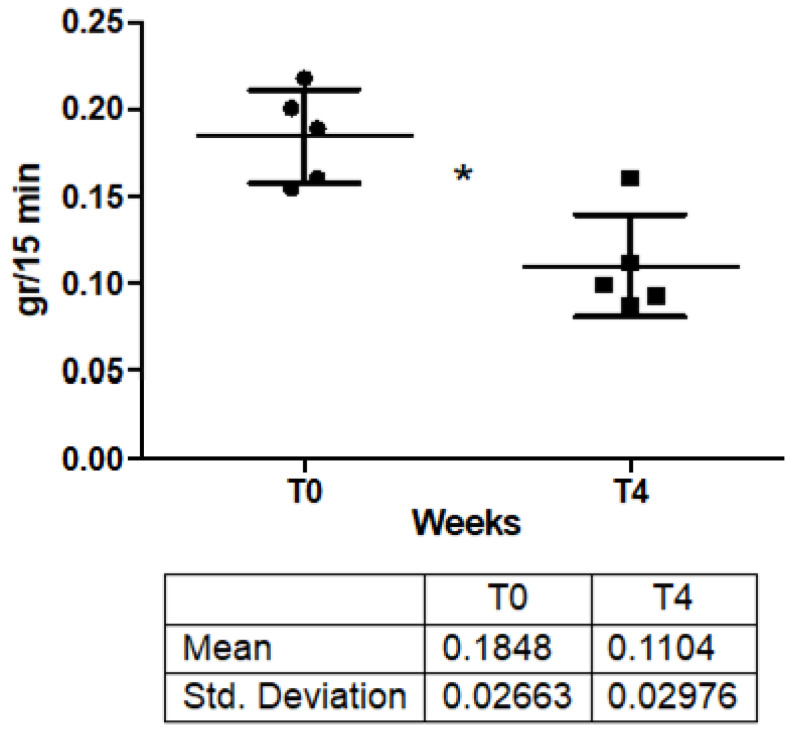
Sweat production before (baseline) and 4 weeks after treatment with IncoBTX-A. * *p* = 0.009. Squares are single values of sweat production for patients after treatment.

**Figure 3 toxins-14-00372-f003:**
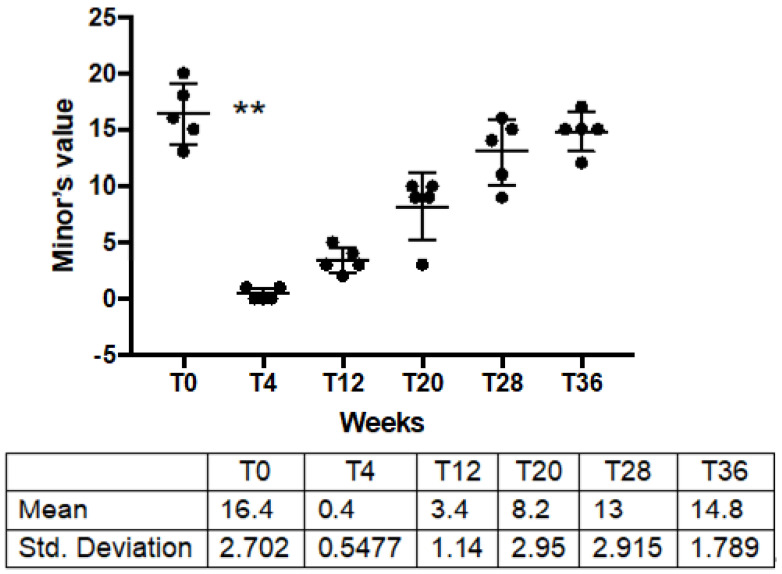
Minor’s iodine starch test before, after IncoBTX-A and at clinical observation point during follow up. ** *p* = 0.004. Solid blocks are single values of sweat production for patients after treatment.

**Figure 4 toxins-14-00372-f004:**
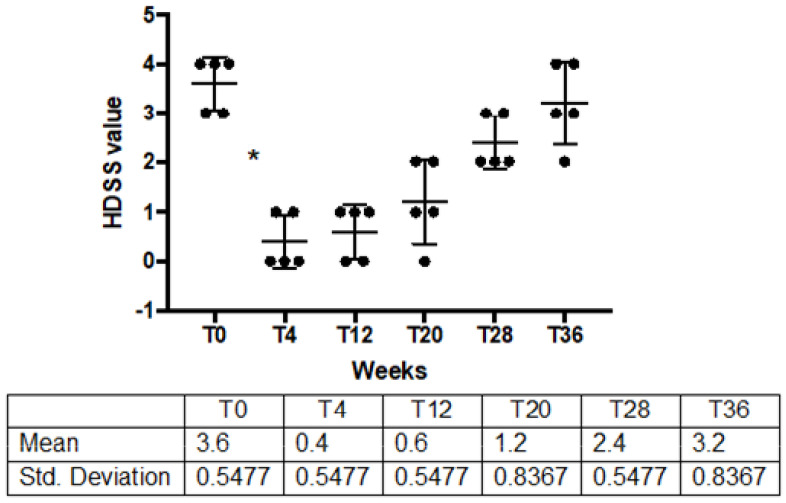
HDSS before, after IncoBTX-A and at clinical observation points during follow up. * *p* = 0.001. Solid black circles are single values of sweat production for patients after treatment.

**Figure 5 toxins-14-00372-f005:**
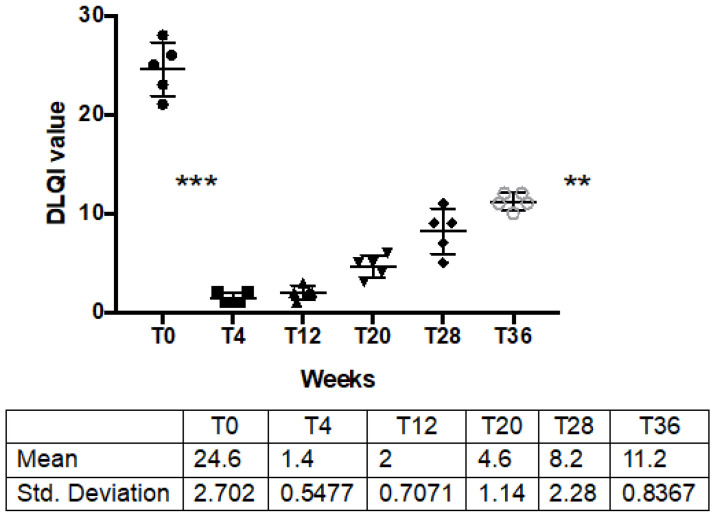
DLQI before, after IncoBTX-A and at clinical observation point during follow up. *** *p* = 0.0007; ** *p* = 0.02. Solid black circles, squares strangles and all other symbols are all single values of DLQI through time-points.

**Table 1 toxins-14-00372-t001:** HH Clinimetric indices before and after treatment with IncoBTX-A.

Pz (*n*)	1	2	3	4	5	*p*
Sweat production T0 (gr/15 min)	0.218	0.201	0.161	0.155	0.819	*p* = 0.009
Sweat production T4 (gr/15 min)	0.093	0.161	0.087	0.112	0.099	
Minor’s iodine starch test T0	20	18	15	16	13	*p* = 0.004
Minor’s iodine starch test T4	0	1	1	0	0	
HDSS T0	4	4	3	3	4	*p* = 0.001
HDSS T4	1	0	0	0	1	
DLQI T0	28	23	25	21	26	*p* = 0.0007
DLQI T4	1	2	2	1	1	

## Data Availability

Not applicabale.

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
