# Peer review of "Botulinum Toxin Type A for Treatment of Forehead Hyperhidrosis: Multicenter Clinical Experience and Review from Literature"

_toxins, 2022, doi:10.3390/toxins14060372_

Round 1

Reviewer 1 Report

Lines 143-144. Claims of less immunogenicity of the mentioned botulinum toxin are widely disputed in the literature, both in articles and at conferences. Indeed, the mass of the injected active ingredient is so negligible that the presence of hemagglutinins in an amount comparable to the mass of the toxin negligibly affects immunogenicity against the background of the denervating effect. Since the authors mentioned are affiliated with the manufacturers of this botulinum toxin, I would suggest that this phrase be preceded by "according to the manufacturer".

Lines 151-152.  There is not a single experiment in the article to support this claim, nor is there any reference to scientific papers that show the lack of the desired effect with the particular ways of applying the selected botulinum toxins. I suggest that this phrase be removed.

Author Response

Reviewer 1)

Lines 143-144. Claims of less immunogenicity of the mentioned botulinum toxin are widely disputed in the literature, both in articles and at conferences. Indeed, the mass of the injected active ingredient is so negligible that the presence of hemagglutinins in an amount comparable to the mass of the toxin negligibly affects immunogenicity against the background of the denervating effect. Since the authors mentioned are affiliated with the manufacturers of this botulinum toxin, I would suggest that this phrase be preceded by "according to the manufacturer". Thank you for bringing to our attention the lack of citations in this aspect. Several investigations have shown the reduced antigenicity of incobotulinum toxin in comparison to other type A toxins. The therapeutic implications of these discrepancies have just been described; we feel that a location as confined as the forehead, proximal to the eye muscles, requires an appropriate choice of toxin in light of the available knowledge. References have been added.

Lines 151-152.  There is not a single experiment in the article to support this claim, nor is there any reference to scientific papers that show the lack of the desired effect with the particular ways of applying the selected botulinum toxins. I suggest that this phrase be removed. Thank you for your insight. It’s true that few investigations have revealed that the spread of ONA and INCO are comparable. Certainly, further clinical research is necessary. References have been added

Reviewer 2 Report

An interesting article. I suggest to shorten the discussion by including a table with the studies published on the effect of different BTX (including the current study) in forehead hyperhidrosis including Authors, Year of publication, type of BTX, design of the study, and main results, including the duration of the possitive effect. In addition, a brief discussion on the differences between the effects of the types of BTX studied and the limitations of the current study should be useful 

Author Response

 (Reviewer 2)

An interesting article. I suggest to shorten the discussion by including a table with the studies published on the effect of different BTX (including the current study) in forehead hyperhidrosis including Authors, Year of publication, type of BTX, design of the study, and main results, including the duration of the positive effect. In addition, a brief discussion on the differences between the effects of the types of BTX studied and the limitations of the current study should be useful. Literature data concerning non-Inco BTX in forehead’s hyperhidrosis has been reported in the discussion section, table has been added and Limitations are now declared.

Reviewer 3 Report

The authors conducted a prospective cohort study to investigate the clinical outcome of patients with forehead hyperhidrosis (FHH) receiving Incobotulinum toxin type A injection. The main conclusion was that the use of IncoBTX-A Injection could reduce the forehead hyperhidrosis. The main drawback was limited patients number (n=5). I have the following comments and suggestions.

  1. Introduction: Please shortly summarize previous literatures about the clinical efficacy of safety of different four botulinum toxin A for treating FHH. A table may be needed.
  2. In addition, some evidences are also needed about clinic outcome about IncoBTX-A for treating patients with different type focal hyperhidrosis. With these evidences, the hypothesis of the study could be proposed reasonably.
  3. Materials and Methods: How to define resistant to antiperspirants?
  4. The sweat production is influenced by the room temperature and humidity. Was every test performed under the similar room temperature and humidity? And what were the temperature and humidity?
  5. Treatment with IncoBTX-A: Why used BTX-A 0.1mL (2mL)? Reference was needed. Was local anesthesia needed before BTX-A injection?
  6. Results: The extent of hyperhidrotic area was restored to baseline at 28weeks and 36 weeks. Any suggestion for the relapsed patients at 36 weeks?
  7. Forehead hyperhidrosis was improved after IncoBTX-A injection. However, whether other area hyperhidrosis would increase?

Author Response

(Reviewer 3)

The authors conducted a prospective cohort study to investigate the clinical outcome of patients with forehead hyperhidrosis (FHH) receiving Incobotulinum toxin type A injection. The main conclusion was that the use of IncoBTX-A Injection could reduce the forehead hyperhidrosis. The main drawback was limited patients number (n=5). I have the following comments and suggestions.

Introduction: Please shortly summarize previous literatures about the clinical efficacy of safety of different four botulinum toxin A for treating FHH. A table may be needed. DONE

  1. In addition, some evidences are also needed about clinic outcome about IncoBTX-A for treating patients with different type focal hyperhidrosis. With these evidences, the hypothesis of the study could be proposed reasonably. Literature data concerning efficacy of IncoBTX-A has been reported in introduction section.

  2. Materials and Methods: How to define resistant to antiperspirants? Resistance to treatment was defined as less than two-points improvement in HDSS from baseline

  3. The sweat production is influenced by the room temperature and humidity. Was every test performed under the similar room temperature and humidity? And what were the temperature and humidity? Since sweat production is influenced by room temperature and humidity, all data were collected under standardized environmental conditions: room temperature ranged from 20 to 22°C, and ambiental humidity from 55 to 60%.

  4. Treatment with IncoBTX-A: Why used BTX-A 0.1mL (2mL)? Reference was needed. Treatment protocol was in accordance with suggestions provided by Ando Y et al. 2021 to treat forehead18 Was local anesthesia needed before BTX-A injection? No local anesthesia was required for treatment

  5. Results: The extent of hyperhidrotic area was restored to baseline at 28weeks and 36 weeks. Any suggestion for the relapsed patients at 36 weeks? The sweat suppression period lasted up to 36 weeks, this result is in line with the mechanism of action of IncoBTX-A which can induce functional chemo-denervation of cholinergic nerve terminations, through molecular disruption of the SNAP- 25 protein. Duration of anhidrosis phase varies according to the individual resynthesis ability of SNAP 25, and in the frontal region data from literature are in agreement with length of time observed by us.

6. Forehead hyperhidrosis was improved after IncoBTX-A injection. However, whether other area hyperhidrosis would increase? No compensatory hyperhidrosis was observed among treated patients.

Reviewer 4 Report

The authors have demonstrated a research on forehead hyperhidrosis treated with botulinum toxin which have been not much described in literatures. They have treated and proved the effect of botulinum toxin effective for forehead hyperhidrosis. The measurement have been demonstrated by gravimetric test, Minor’s iodine starch test, hyperhidrosis disease severity scale (HDSS) and dermatologic quality of life index (DQLI). They have followed the significant anhidrotic effect over 4 weeks and proved to be effective until approximately 36 weeks. They have reported minor side effect of mild transient bilateral ptosis. The article have been well organized with sorting reviews and their own experiments.

Abstract

Line 14 : Is the abbreviation for dermatologic quality of life index (DQLI? Or DLQI?) please have the correct term.

Methods: How was the dose of the botulinum toxin units have been used. How was the mixture rate with normal saline. Did the injection points differ from one patient to another? Please specify

I would like to give a minor revision for author’s great work to the subject.

Author Response

(Reviewer 4)

The authors have demonstrated a research on forehead hyperhidrosis treated with botulinum toxin which have been not much described in literatures. They have treated and proved the effect of botulinum toxin effective for forehead hyperhidrosis. The measurement have been demonstrated by gravimetric test, Minor’s iodine starch test, hyperhidrosis disease severity scale (HDSS) and dermatologic quality of life index (DQLI). They have followed the significant anhidrotic effect over 4 weeks and proved to be effective until approximately 36 weeks. They have reported minor side effect of mild transient bilateral ptosis. The article have been well organized with sorting reviews and their own experiments.

Abstract
Line 14 : Is the abbreviation for dermatologic quality of life index (DQLI? Or DLQI?) please have the correct term. Abbreviation is for Dermatology Life Quality Index (DLQI)
Methods: How was the dose of the botulinum toxin units have been used. How was the mixture rate with normal saline. Did the injection points differ from one patient to another? Please specify

Treatment protocol was in accordance with suggestions provided by Ando Y et al. 2021 to treat forehead18. All patients received Incobotulinumtoxin type A (IncoBTX-A) at a fixed dosage per cm2. A nurse external to the study diluted every vial of lyophilised incobot-ulinum toxin type A containing 100 U (Xeomin®, Merz Pharma, GmbH & Co KGaA) in 5 mL sterile 0.9% saline solution. Following this dilution procedure, 1 mL of reconstituted product contained 20 U of IncoBTX-A, and 0.10 mL contained 2U of them. A reference grid with square areas of 2.25 cm2 was drawn in the forehead, the in-tra-cutaneous injection of IncoBTX-A 0.10 mL (2U) was given by the physician in the central part of each square. Following this procedure, injection points did not differ from one patient to another.

I would like to give a minor revision for author’s great work to the subject.